# Impacts of Imidacloprid and Flupyradifurone Insecticides on the Gut Microbiota of *Bombus terrestris*

Qingchao Zhang [1,2], Qinglin Wang [2], Yifan Zhai [1,3,4], Hao Zheng [1,3,4,*] and Xiaofei Wang [2,*]

1 Institute of Plant Protection, Shandong Academy of Agricultural Sciences, Jinan 250100, China; zhangqingchaozy@163.com (Q.Z.); zyifan@tom.com (Y.Z.)
2 College of Food Science and Nutritional Engineering, China Agricultural University, Beijing 100083, China
3 Key Laboratory of Natural Enemies Insects, Ministry of Agriculture and Rural Affairs, Jinan 250100, China
4 Shandong Provincial Engineering Technology Research Center on Biocontrol of Crops Diseases and In-sect Pests, Jinan 250100, China
* Correspondence: hao.zheng@cau.edu.cn (H.Z.); xiaofei.wang@cau.edu.cn (X.W.)

**Abstract:** Bumblebees are important pollinators for crops and wild flowering plants. Various pesticides have threatened the abundance and diversity of bumblebees. In addition to direct sublethal effects, pesticides may alter the gut microbial communities of bees. Imidacloprid and flupyradifurone insecticides both bind to the nicotinic acetylcholine receptor. However, the latter was assumed to be harmless for honeybees and can even be applied to flowering crops. In this study, we assessed the impacts of these two pesticides on queenless microcolonies and the gut microbiota of *Bombus terrestris*. We found that 10 μg/L imidacloprid significantly impeded syrup consumption, and postponed the egg-laying period, larvae, and pupae development. It decreased the relative abundance of the bumblebee-specific symbionts, *Apibacter* and *Lactobacillus* Firm-5. On the contrary, 10 μg/L flupyradifurone did not reduce syrup consumption, block larvae and pupae development in bumblebees. Although no significant phenotypes were observed, PICRUST revealed that flupyradifurone suppressed pathways, involving carbohydrate metabolism, nucleotide metabolism, translation, and membrane transport. Our findings suggest the appropriate use of this new pesticide may be considered safe for bumblebees, but the underlying mechanism warrants further investigation.

**Keywords:** *B. terrestris*; imidacloprid; flupyradifurone; microcolony; gut microbiota

## 1. Introduction

Bumblebees (*Bombus* genus) play a vital role as efficient pollinators in agricultural and natural systems. Specifically, several bumblebee species, such as *Bombus impatiens* and *Bombus terrestris,* have been domesticated for commercial pollination service, for both glasshouse and open-field crops [1]. However, over recent years, populations of bumblebees have continued to decline, on a global scale [2]. Multiple factors may contribute to the decline of the bumblebee population, such as habitat loss, parasites and diseases, invasive species, and pesticide exposure [3–7]. Both laboratory and field studies demonstrated the negative impacts of pesticides on bumblebee reproduction, colony development, and behavior [8–10]. Currently, neonicotinoids are the most widely used pesticide class, resulting in the declined population of wild bees and other insects [11]. Concentrations of pesticide residues in pollen and nectar vary considerably, while average maximum values are around 2 ppb for nectar and 6 ppb for pollen. Therefore, assessments on the risks of the exposure of bumblebees to insecticide compounds are necessary.

Imidacloprid [1-(6-chloro-3-pyridylmethyl)-2nitroimino-imidazolidine] was the first neonicotinoid in widespread use [12,13]. Imidacloprid can act as an agonist of nicotinic acetylcholine receptors (nAChR), which play a significant role in mediating fast excitatory synaptic transmission in the central nervous system of insects [14]. Imidacloprid can be

absorbed by plants and transported throughout the plant's vascular system. With the detection of imidacloprid residues in pollen, nectar, and guttation fluids, worker honeybees and other insects face threats, since they suck the plant fluid and juices for nutrition [15]. Recent studies have shown that sublethal doses of imidacloprid cause changes in reproduction and impair the foraging behavior of honeybees [16]. Concentrations of imidacloprid as low as 1 µg/L can cause reduced foraging motivation in *B. terrestris* [17]. In addition, acute imidacloprid exposure (10 ppb, 10 min) altered mitochondrial function in the bumble flight muscle and brain [18]. Concerning the adverse physiological and ecological effects, the field use of imidacloprid has been restricted by the European Union [19]. Hence, there is an urgent need to find alternative molecules for effective pesticides.

Flupyradifurone (4-[(2,2-difluoroethyl) amino]-2(5 H)-furanone) is a novel pesticide (commercial product Sivanto ®), launched by Bayer AG (Monheim am Rhein, Germany) [20]. Compared to imidacloprid, flupyradifurone also acts as a reversible agonist on insect nAchR, while at a different site of action [21]. Flupyradifurone has been claimed to be "slightly toxic" to organisms in the environment, while it may evoke bees' motor disabilities and disturb normal motor behavior [20]. Flupyradifurone is nontoxic to adult bees on an acute contact exposure basis [22], while chronic exposure (around 1.5 µg/L) can lead to premature foraging in honeybees [23]. Moreover, long-term field-realistic exposure of flupyradifurone may impair honeybee behavior and survival [24]. So far, there is little evidence in the literature to show how flupyradifurone may affect the survival, development, and foraging activity of bumblebees.

In addition to the sublethal effect and development impairment on colonies, insecticides have significant adverse effects on pollinating bees' gut bacterial composition and diversity [25–27]. Exposure to imidacloprid affected the gut microbiota composition of honeybees, and the abundance of *Lactobacillus* Firm-5 was significantly decreased [28].

Like honeybees, bumblebees have a relatively simple, but host-specific, gut microbiota [29]. There are mainly five bacterial genera forming the core microbiota: *Snodgrassella*, *Gilliamella*, *Lactobacillus* Firm-4 (genus *Bombilactobacillus* now), *Lactobacillus* Firm-5, and *Bifidobacterium* [30]. However, *Apibacter* was recently identified as a genus of bee-specific bacteria that is only prevalent in the guts of *Apis cerana*, *Apis dorsata*, and *Bombus* species, but is sporadically observed in *Apis mellifera* [29,31]. The host-restricted microbiota associated with honey- and bumblebees play important roles in health, such as nutrient metabolism, immune regulation, and pathogen resistance [32,33]. Detrimental effects on gut microbiota in honeybees of dietary neonicotinoids have already been demonstrated, but the effects on bumblebees are unclear [34]. Therefore, it is vital to understand the impacts of pesticides on bumblebee intestinal microbes, specifically the potential effects of different insecticides.

The queenless microcolony of bumblebees is a group of workers (3–6 individuals) isolated in an environment without a mature queen (Figure 1B) [35]. Microcolonies have been used to assess the impacts of pesticide exposure in a controllable and repeatable way within the laboratory [36]. In this study, we utilized the microcolony model to evaluate the effects of two different pesticides, imidacloprid and flupyradifurone, on the development of a bumblebee colony. Further, high-throughput 16S rRNA gene sequencing was used to identify the microbial communities associated with worker adult bumblebees after chronic exposure to imidacloprid or flupyradifurone.

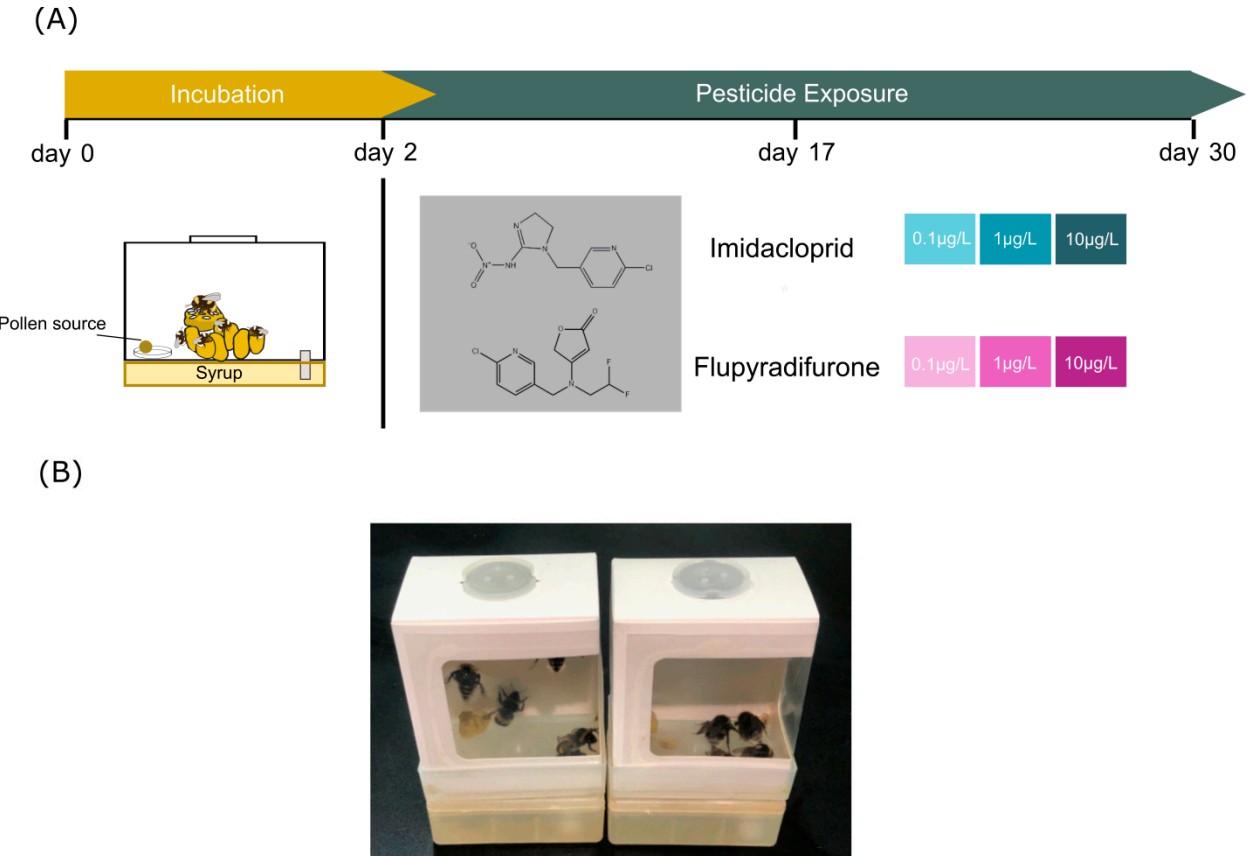

**Figure 1.** Scheme of experiment. (**A**) Bumblebees and queenless microcolonies were generated from pupae. They were fed with imidacloprid and flupyradifurone, separately, for 15 days prior to sacrifice and harvest of gut samples. (**B**) Capture of microcolony in the lab.

## 2. Materials and Methods

### 2.1. Bumblebee Microcolony and Insecticides Treatment

*B. terrestris* was obtained from Shandong Lubao Technology Development Co. Ltd (Jinan, China). About 70 beehives were collected. Unhatched black pupae that would emerge into adult workers were picked and placed in a small white box (75 × 50 × 90 mm) on top of the original beehive. The small white box was checked twice every day, and the newborn worker bees were marked and transferred directly into a bigger yellow hive (150 × 200 × 200 mm). Then the collected newborn individuals from the yellow boxes were mixed and replaced into the hives, each with 15–20 bumblebee individuals. All hives were maintained in an environmental chamber in continuous darkness at 65 ± 5% relative humidity and 29 ± 0.5 °C for the duration of the whole experiment.

According to Klinger et al. [36], we used microcolonies of *B. terrestris* to assess pesticide risk by checking the development and food consumption. To initiate microcolonies, worker bees were divided into five groups, then transferred to microcolony cages (120 × 120 × 100 mm) (Figure 1B). We placed a circular protrusion in the hive to encourage worker bees to lay eggs [36]. Microcolonies were maintained at 29 ± 0.5 °C and 65 ± 5% relative humidity.

The microcolony performance can be evaluated by the development of larvae and pupae, which provide insights into the toxicity against the offspring exposed to insecticides [36]. The microcolonies developed into larvae within one week after laying eggs and staged to pupae in the next seven days [36]. The microcolonies were terminated after 28 days to ensure that individuals could develop into the pupal stage [37,38]. Bumblebees used in all experiments were generated from pupae that emerged in the lab. Newly-emerged bees were treated with pesticides by oral feeding for 15 days, while the micro-

colonies were treated with insecticides for 28 days to investigate the effects on colony development. Imidacloprid and flupyradifurone nominal concentrations in syrup were 0.1, 1, and 10 μg/L, abbreviated as IL-0.1, IL-1, IL-10, F-0.1, F-1, and F-10 in the text, while CK means the control group without any insecticide supplementation. Sugar syrup (50%, *w/w*) was made with pure cane sugar and distilled water. The syrup was replaced every three days, and control microcolonies were provided with filter-sterilized sugar syrup. Imidacloprid was obtained from Shandong United Pesticide Industry Co., Ltd. (Jinan, China), and the pesticide registration certificate number is PD20131368, while flupyradifurone was obtained from Bayer AG (pesticide registration certificate number: PD20184006). To analyze microcolony development, diet consumption, number, and weight of larvae and pupae were recorded.

### 2.2. Gut Microbiota

To study the effect of imidacloprid and flupyradifurone on the bumblebee gut microbiome, surviving worker bees were collected on day 17 from each colony. Whole guts were dissected using fine-tipped forceps and were homogenized with a plastic pestle. The gut DNA was extracted using the cetyltrimethylammonium bromide (CTAB) buffer method. DNA quality and quantity were assessed by visualization at 260 nm/280 nm and 260 nm/230 nm, respectively, using a Nanodrop ND 2000.

To determine the indirect effects of these two pesticides on the size and composition of the gut microbiome, the guts of bumblebees were sampled from each group, and relative abundances of gut bacteria were assessed using deep amplicon sequencing of the V4 region of the bacterial 16S rRNA gene. The microbiota were profiled by sequencing the V4 region of the 16S ribosomal RNA (rRNA) gene. Primer pairs 515F (5′-GTGCCAGCMGCCGCGGTAA-3′) and 806R (5′-GGACTACHVGGGTWTCTAAT-3′) were used to amplify the V4 region. PCR reactions were carried out with 15 μL of Phusion® High-Fidelity PCR Master Mix (New England Biolabs, Ipswich, UK), 0.2 μmol/L each of forward and reverse primer 10 ng template DNA. The PCR cycle was 95 °C (30 s) followed by 40 cycles of 95 °C (5 s) and 60 °C (30 s). The reaction was performed on the 7500 Real-Time PCR System (Thermo Fisher, Shanghai, China).

Bioinformatic analysis was performed using the QIIME2 pipeline (http://qiime.org/) [39] (accessed 5 Jun 2020) and mothur software (https://mothur.org/) [40] (accessed 10 Jun 2020). Principal coordinates analysis (PCoA) and alpha diversity indices were visualized in R (version 3.4.1). Beta diversity was calculated and visualized by generating principal coordinate plots. Functional prediction of gut microbiota was made by PICRUSt analysis of the OTUs obtained from the Greengenes reference database [41].

### 2.3. Ethics Statement

The present experiment was conducted at the Shandong Institute of Plant Protection Jinan, China. The research protocol was approved by the Animal Ethics of Shandong Academy of Agricultural Sciences under ethic approval number SAAS-2022LL-01.

## 3. Results

To determine how these two pesticides impact bumblebee health, we used queenless bumblebee microcolonies, provisioned with different concentrations. Since the pollen from imidacloprid-treated crops generally contains residues of neonicotinoids, ranging from 1 μg/kg to 3 μg/kg [42–44], we set the concentration of imidacloprid and flupyradifurone at 0.1, 1, and 10 μg/L to investigate whether exposure to insecticides affects bumblebee performance over similar levels of field-realistic exposure.

### 3.1. Chronic Exposure to Imidacloprid Caused Detrimental Effects on Bumblebees

We first examined the pesticide-induced impact on the food intake of *B. terrestris*. There was no significant difference in syrup consumption during the first week (Figure 2A). However, a substantial decrease in food consumption was evident only in microcolonies

exposed at the highest dosage (10 μg/L) of imidacloprid, to the extent of 15% ($p < 0.05$), while no variation was detected among microcolonies exposed to 1 μg/L of imidacloprid. Separation from the queen stimulates one of the workers (usually the one with the most developed ovaries) to establish dominance and begin laying eggs. The egg-laying time was an important indicator of microcolony development [36]. All bumblebees spawned within five days (Figure 2B), indicating that the queenless microcolony was normal and stable for generating toxicity data. A low dosage (1 μg/L) of imidacloprid did not change the spawning time of the dominant worker. In comparison, higher concentration (10 μg/L) exposure postponed the spawning activity for approximately one week for one-third of the worker bees ($p < 0.05$). Overall, a high dose of imidacloprid treatment reduced the diet consumption of the whole colony and postponed the microcolony's egg-laying period, while no apparent effects were observed for flupyradifurone.

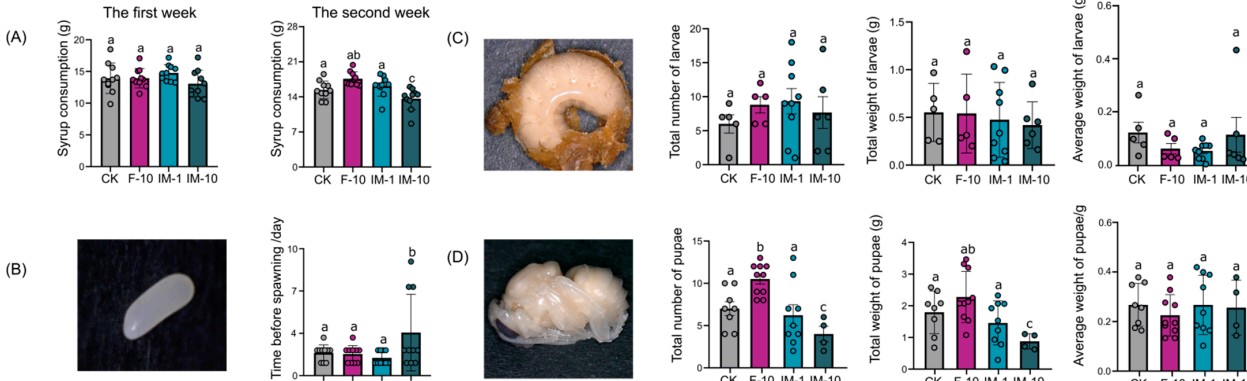

**Figure 2.** Effect of pesticide exposure on bumblebee larval and pupal development. (**A**) Syrup consumption in *B. terrestris* worker microcolonies (*n* = 40) after two weeks of pesticide exposure treatments. (**B**) Time of first egg deposition in *B. terrestris* worker microcolonies. (**C**) The total number, total weight and average weight of larvae for different pesticide treatments. (**D**) The total number, total weight and average weight of pupae for different pesticide treatments. Significant differences between groups were determined by one-way ANOVA with Tukey's multiple comparisons test. Letters a, b and c indicate statistically significant differences between treatments. (CK, F-10, IM-1 and IM-10 represent control, 10 μg/L flupyradifurone, 1 and 10 μg/L imidacloprid exposure).

No significant differences were shown in the total number and weight of larvae for both 1 and 10 μg/L imidacloprid exposure (Figure 2C). Nevertheless, during the following pupae stage, the total weight and number of pupae obviously decreased in the treated group with 10 μg/L imidacloprid, compared to other concentrations ($p < 0.05$). The average weight of pupae under imidacloprid exposure remained similar to the control group in the microcolonies (Figure 2D).

*3.2. Chronic Exposure to Flupyradifurone Shows No Detrimental Effects in Food Uptake and Development*

To observe whether flupyradifurone displayed detrimental effects, as imidacloprid, bumblebees were fed with a 10μg/L flupyradifurone diet. Intriguingly, this treatment showed resistance to syrup consumption (Figure 2A), which was not significantly different from the CK group. Using *B. terrestris* microcolonies, we then characterized the effects of flupyradifurone on the production of the egg, larvae, and pupae offspring. Initial egg deposition under 10 μg/L flupyradifurone treatment occurred within four days, similar to that of the CK group (Figure 2B). This situation was different from that observed among imidacloprid exposure with a seven-day before spawning, although the same concentration was still capable of laying eggs.

The total number and weight of larvae tended to be similar in CK compared with bumblebees fed with 10 μg/L flupyradifurone (Figure 2C). Consistent with previous reports,

low-level chronic neonicotinoid exposure on bumblebees did not affect the total number of larvae produced by workers in microcolonies [45]. Furthermore, 10 µg/L flupyradifurone exposure slightly increased the total number of pupae, while keeping the same level of total and average weight, indicating no potential effects on pupal development (Figure 2D). Therefore, the appropriate use of this pesticide is considered safe for bumblebees, at least for the queenless microcolony studied here. Similarly, flupyradifurone (Sivanto, Bayer AG, Monheim am Rhein, Germany) was reported to be harmless for honeybees' taste and cognition behaviors under field conditions [21].

### 3.3. Both Imidacloprid and Flupyradifurone Exposure Perturbed the Profiles of Gut Microbiota

Exposure to imidacloprid disturbed the microbial community richness, as indicated by the Chao1 index and Shannon's H index, especially IM-10 treatment, with a sharp decrease (Figure 3A,B). The principal coordinate analysis of weighted UniFrac showed the gut community compositions of imidacloprid-exposed bumblebees differed from the control group, especially IM-10 treatment. Although different concentrations of imidacloprid changed the gut composition, all core species of the bumblebee gut were present with *Lactobacillus* Firm-5, *Snodgrassella*, *Gilliamella*, *Apibacter,* and *Bifidobacterium*. The effects of imidacloprid exposure on the bumblebee gut microbiome were more prominent at concentration 10 µg/L, relative to control, which also exhibited more severe compositional shifts (Figure 3D). Specifically, the relative abundances of two dominant gut bacteria, *Apibacter* and *Lactobacillus* Firm-5 (Figure 3E,F), were both decreased ($p < 0.05$). This perturbation may contribute to metabolic homeostasis.

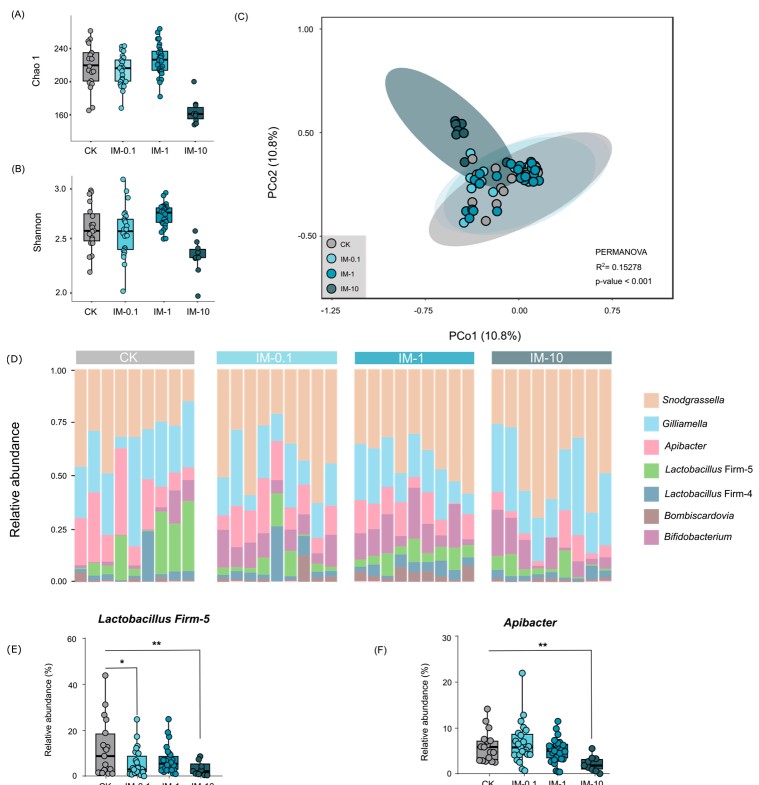

**Figure 3.** Gut microbiota compositions of the bumblebees under chronic exposure of imidacloprid. (**A**) α-Diversity at genus level estimated by Chao1 richness index. (**B**) Shannon diversity estimator. (**C**) Principal coordinate analysis of gut community composition in control and imidacloprid-treated bumblebees. (**D**) Bar graph of bacterial abundance at the genus level. (**E**) The relative abundance of *Lactobacillus* Firm-5 in the bumblebee gut. (**F**) The relative abundance of *Apibacter* in the bumblebee gut. (*$p < 0.05$, **$p < 0.01$) (CK: $n = 19$, IM-0.1: $n = 23$, IM-1: $n = 28$, IM-10: $n = 9$) CK, IM-0.1, IM-1 and IM-10 represent control, 0.1, 1, and 10 µg/L imidacloprid treatment.

To examine whether changes in the gut microbiota mediated differences in metabolic health between these two pesticide groups, 16S rRNA gene sequencing was also performed on bumblebees fed with flupyradifurone. According to the indices of microbial community richness, Chao1, and Shannon index, no significant changes in gut microbiota composition were found among flupyradifurone-treated bumblebees (Figure 4A,B). PCOA analysis revealed no distinct clustering of flupyradifurone-treated and control groups, based on microbiota composition (Figure 4C), contrary to that of imidacloprid exposure. As described in the previous section, assessment of gut microbiomes identified all core gut taxa, *Lactobacillus* Firm-5, *Snodgrassella*, *Gilliamella*, *Apibacter*, and *Bifidobacterium*, in flupyradifurone treatment groups (Figure 4), showing that flupyradifurone did not eliminate colonization of any core member. Different from the imidacloprid experiment, the relative abundance of *Apibacter* and *Lactobacillus* Firm-5 kept a similar level, and these bacteria may be predicted to be unaffected by flupyradifurone exposure. However, a significant increase in abundance was observed for *Bifidobacterium* in bumblebees treated with 10 µg/L flupyradifurone, similar to that of bees exposed to the insecticide nitenpyram [46].

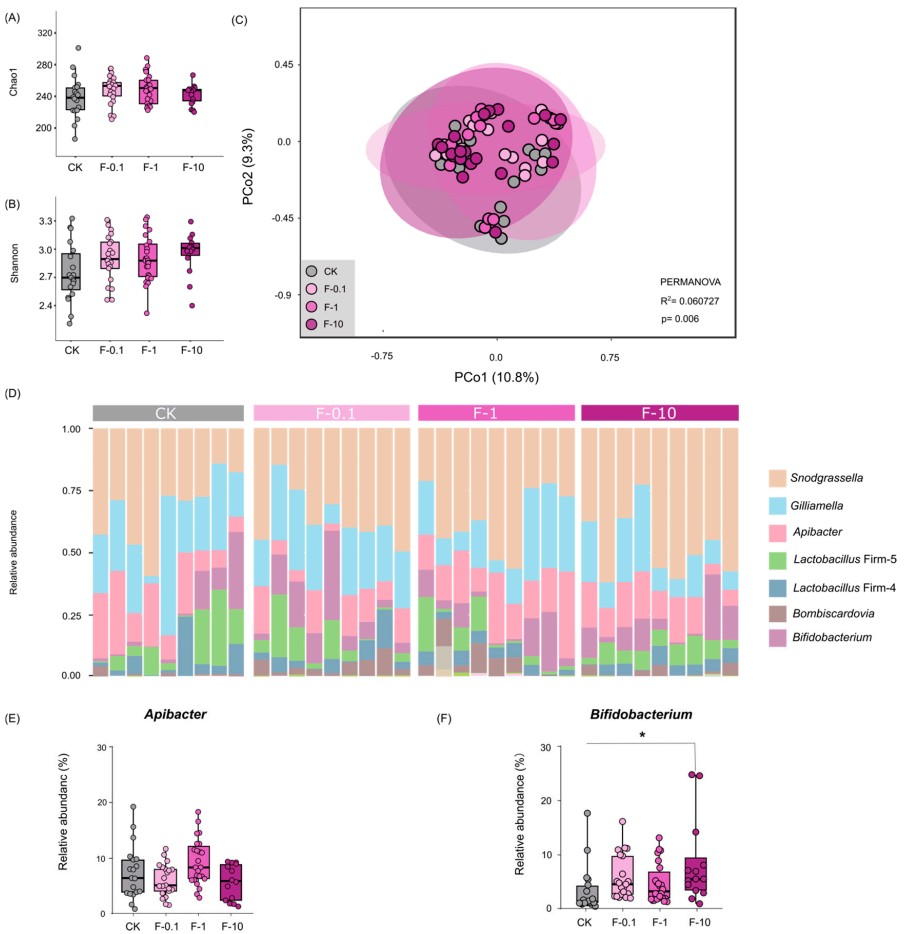

**Figure 4.** Gut microbiota compositions of the bumblebees under chronic exposure of flupyradifurone. (**A**) α-Diversity at genus level estimated by Chao1 richness index. (**B**) Shannon diversity estimator. (**C**) Principal coordinate analysis of gut community composition in control and flupyradifurone-treated bumblebees. (**D**) Bar graph of bacterial abundance at the genus level. (**E**) The relative abundance of *Apibacter* in the bee gut. (**F**) The relative abundance of *Bifidobacterium* in the bee gut. (\*$p < 0.05$) (CK: $n = 19$, F-0.1: $n = 21$, F-1: $n = 24$, IM-10: $n = 14$) CK, F-0.1, F-1 and F-10 represent control, 0.1, 1, and 10 µg/L flupyradifurone treatment.

### 3.4. Effect of Imidacloprid and Flupyradifurone on the Potential Functions of the Gut Microbiome

To understand the metabolic potential of gut microbes from pesticide exposure, functional profiles of 16S rRNA data were inferred using PICRUSt analysis. PCOA analysis revealed distinct clusters between CK and the imidacloprid-treated group, indicating that imidacloprid exposure significantly influenced microbial metabolism (Figure 5A). Moreover, the functional profile changed with different imidacloprid concentrations (Figure 5B). Shifts in the functional categories related to metabolic pathways were further investigated statistically (Figure 5C). Specifically, imidacloprid exposure significantly suppressed carbohydrate metabolism and membrane transport, and pathways involving nucleotide metabolism and translation were also weakened in IM groups. In contrast, the metabolism of xenobiotics, amino acids, energy, cofactors, and vitamins were enhanced (Figure 5C).

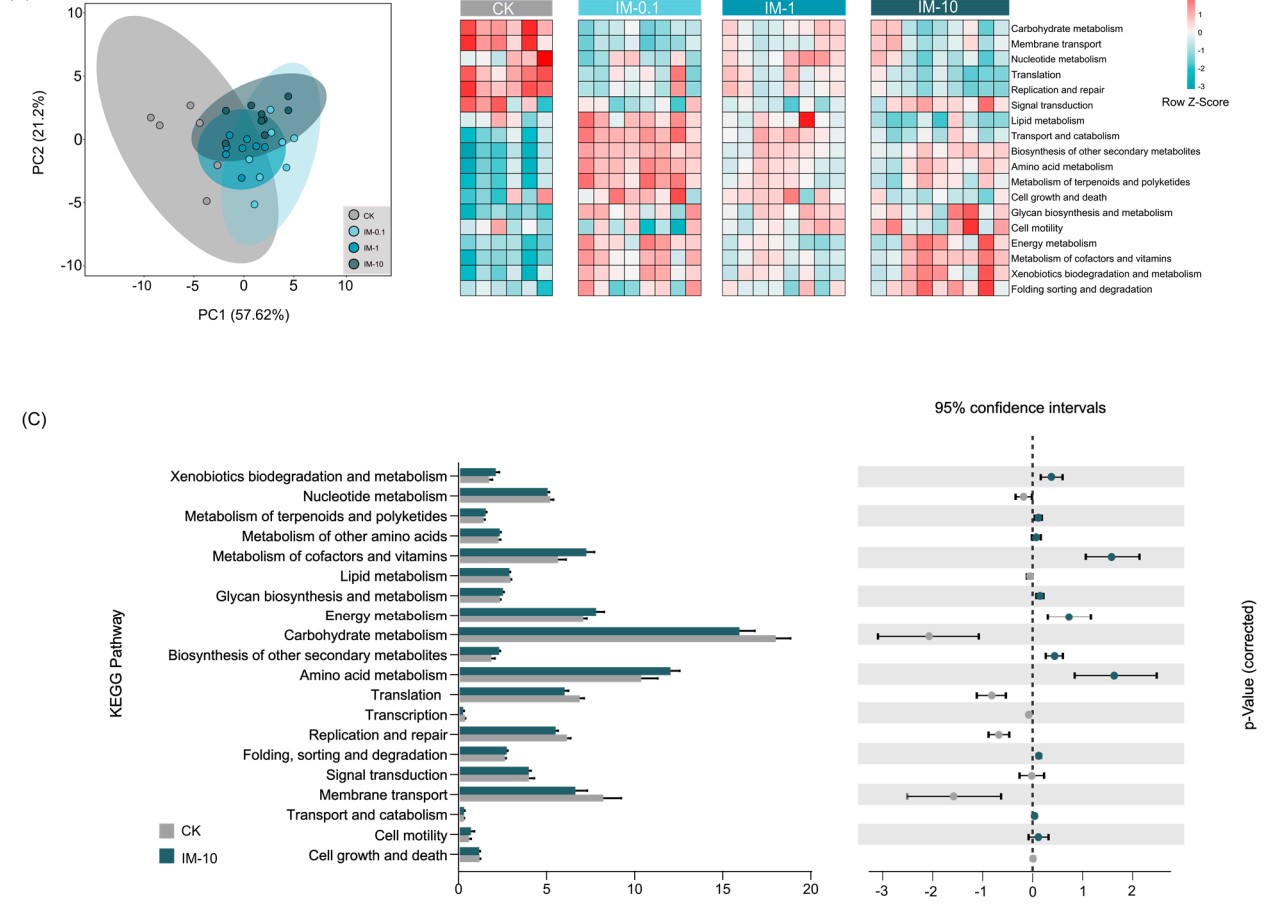

**Figure 5.** Functional profiles of the gut microbiota between CK and imidacloprid treated bumblebees. (**A**) PCoA plot describing functional inferences (PICRUSt) of bacterial communities across imidacloprid treatments. (**B**) Heatmap displaying the differentially enriched KEGG pathways (Level 2) prediction by PICRUSt across different concentration treatment groups. (**C**) Bar plots showing the relative abundance of KEGG pathways prediction by PICRUSt and the difference between CK and IM-10. CK, IM-0.1, IM-1 and IM-10 represent control, 0.1, 1, and 10 µg/L imidacloprid treatment.

Fascinatingly, the functional profile of the gut microbiota suggested that flupyradifurone may not be as "safe" as was previously documented. A similar pattern was observed between the predicted functional categories of flupyradifurone-exposed and control bumblebees, compared to their imidacloprid counterparts. Although no significant changes were discovered among the structure or composition of the gut community after chronic

flupyradifurone exposure, PICRUSt-generated PCOA analysis revealed distinct clusters between control and flupyradifurone groups, which was in accordance with the results of imidacloprid exposure (Figure 6A). By analogy with imidacloprid treatments, carbohydrate and amino acid metabolism were down- and upregulated, respectively. Similarly, flupyradifurone exposure suppressed membrane transport, metabolism of nucleotide, and translation, while enhanced metabolism of energy, secondary metabolites, cofactors, and vitamins (Figure 6B).

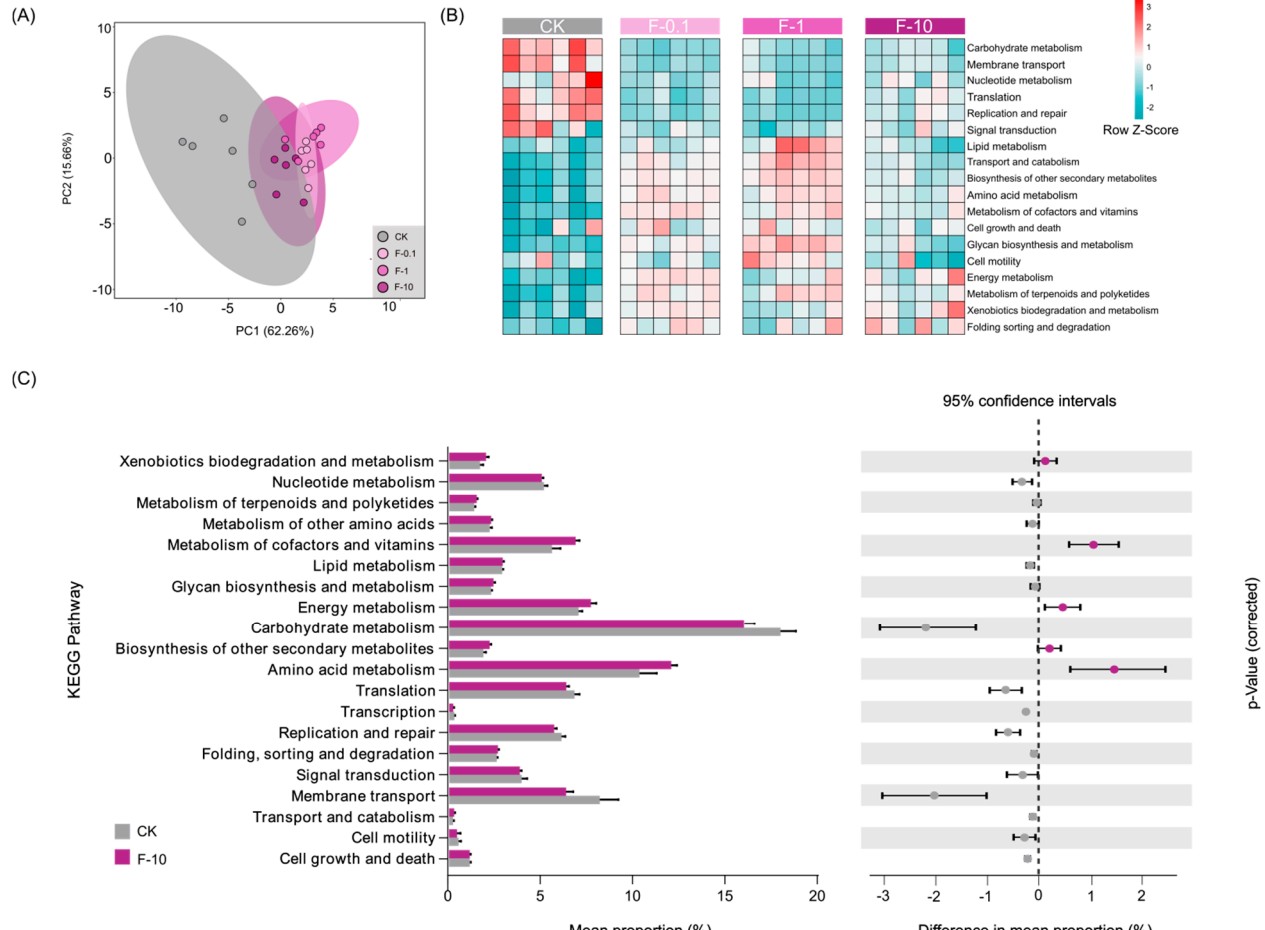

**Figure 6.** Functional profiles of the gut microbiota are distinguishable between CK and flupyradifurone-treated bumblebees. (**A**) PCA plot describing functional inferences (PICRUSt) of bacterial communities across flupyradifurone treatments. (**B**) Heatmap displaying the differentially enriched KEGG pathways (Level 2) prediction by PICRUSt across different concentration treatment groups. (**C**) Bar plots showing the relative abundance of KEGG pathways prediction by PICRUSt and the difference between CK and F-10.

## 4. Discussion

Our data indicated that imidacloprid exposure, significantly, resulted in detrimental effects, as 10 µg/L imidacloprid reduced the syrup consumption of the microcolonies under laboratory conditions. As in the previous study, Laycock reported that 10 µg/L imidacloprid decreased feeding on syrup, probably due to a repellent or antifeedant effect to diminish the ability or demand to feed [47,48]. Other than food consumption, 10 µg/L imidacloprid also reduced the number of larvae and pupae and postponed the time to lay eggs, indicating that few larvae survived long enough to begin pupation. Taken together, feeding with 10 µg/L imidacloprid-contaminated syrup resulted in total larvae weight decline and explained the observed reductions in pupae production. Consistently,

10μg/L imidacloprid-treated *B. terrestris* delayed the time of nest construction and the initiation of egg-laying and reduced larval production by 43% [49,50]. Similarly, 10 μg/kg thiamethoxam, one of the most commonly used nitro-substituted neonicotinoid insecticides, was capable of reducing syrup feeding and delayed nest building activity for queenless microcolonies of *B. terrestris* worker bees [51]. Collectively, the data suggest that imidacloprid may have a delayed, lethal effect on bumblebee microcolonies, with chronic exposure to imidacloprid under artificial conditions.

However, flupyradifurone treatment seemed relatively safe for bumblebees in this study. Concerning syrup consumption, there was no significant difference between the control and flupyradifurone-treated groups. Moreover, dietary flupyradifurone exposure did not affect the larval time, pupal time, or a combination of the two. Therefore, chronic exposure to flupyradifurone was considered relatively safe for bumblebees compared to imidacloprid. As for honeybee experiments, the 4 ppm flupyradifurone-treated group consumed 16% less nectar [52,53]. A previous study also reported that there were no negative effects on the larvae development and foraging activity of honeybees with 4 ppm flupyradifurone [54]. Due to the unique binding mode of flupyradifurone to nAChR, flupyradifurone is considered to have low-toxicity to the honeybee [21].

It has been well established that gut microbes can affect host nutrition, weight gain, endocrine signaling, immune function, and pathogen resistance [55], while perturbation of the microbiota can reduce host fitness. In our study, 10 μg/L imidacloprid significantly perturbed the gut microbiota community structure, shown in both α and β diversity. The relatively different effects of imidacloprid treatment at concentration 10 μg/L on the microbiota composition are unexplained but may reflect developments of dose dependence. Consequently, it caused a significant decrease in important species in the relative abundance of *Lactobacillus* Firm-5 and *Apibacter*. It was documented that imidacloprid exposure to *Drosophila melanogaster* also significantly increased the abundance of *Acetobacter* and *Lactobacillus* genera, supporting the potential toxicity of neonicotinoid pesticides targeting gut microbiota [56]. Moreover, exposure to imidacloprid may interact with the immune deficiency pathway, leading to a loss of microbial regulation, as exemplified by a compositional shift on dominant microbiota members [56]. The relative abundance of *Lactobacillus* Firm-5 decreased, which may suppress carbohydrate metabolism, since it functionally degrades flavonoid glycosides to simple sugars and organic acids further [57,58]. Moreover, *Lactobacillus* Firm-5 possesses numerous phosphotransferase systems involved in the uptake of sugars [59]. *Apibacter* are present for the de novo synthesis of all proteinogenic amino acids, except methionine, for which there is an encoded transporter [60]. Despite the general reduction in the *Apibacter* genome, amino acid metabolism was retained, which is putatively beneficial to the host [61]. In summary, imidacloprid exposure caused significant changes to the structure profile of the *B. terrestris* gut community.

Nevertheless, flupyradifurone exposure only slightly changed the gut microbial community structural richness, except for the significant increase in *Bifidobacterium* in the F-10 group. This change may refer to an adaption of the microbial community, with the probiotic species becoming more abundant to counteract the chronic effects of the insecticide. *Bifidobacterium* has been documented as an important degrader of hemicellulose and pectin for honeybees with strain-level diversity, in gene repertoires linked to polysaccharide digestion [32]. Anderson reported that *Bifidobacterium* possessed catalase, peroxidase, superoxide dismutase, and respiratory chain enzymes, indicative of oxidative metabolism [62]. The perturbation of the gut microbiota demonstrated their potentially nutritional roles in detoxifying molecules in food.

Although the major components of microbiota were not altered, the abundance of several core gut members and the potential functional profiles of the gut microbiota were disturbed. The key result emerging from our work is that ingestion of both imidacloprid and flupyradifurone, at environmentally realistic levels, substantively perturbed the functional profile of bumblebees' gut microbiota. These two pesticides significantly downregulated carbohydrate metabolism and upregulated energy metabolism. The decrease in carbohy-

drate metabolism may account for the increase in energy consumption to supply energy for the detoxification of pesticides [63]. This functional regulation of gut microbiota may confront the nAChR-mediated effects in a carbohydrate-mitochondrial detoxification network for imidacloprid and flupyradifurone [64]. These functional profiles of gut microbes suggest the underlying mechanism of flupyradifurone warrants further investigation, despite its "safe" appearance on bumblebees. Moreover, amino acid metabolism was significantly enhanced under imidacloprid and flupyradifurone exposure. This was reported previously in mice, that neonicotinoid insecticides cause amino acid metabolism disorders, with increases in branched chain amino acids and phenylalanine [65]. It is worth noting that the gut microbiota can synthesize these essential amino acids. However, this information was predicted by bioinformatics analysis; functional metabolic/transcriptional changes in response to insecticide exposure need further measurement and explanation. Other than these obvious changes, the pathway of xenobiotics biodegradation and metabolism was also disturbed. Previous studies have shown that honeybee gut symbiont could contribute to bee health by modifying the host xenobiotics detoxification pathways, and cytochrome P450-mediated detoxification contributes to xenobiotic tolerance in many insects [66]. The upregulation in the metabolism of xenobiotics by cytochrome P450 is consistent with their increased exposure to xenobiotic pesticides, compared to control groups. The significantly elevated expressions of xenobiotic biodegradation and metabolism suggest that the gut microbiota of bumblebees may provide the first line of defense against dietary insecticides, for both imidacloprid and flupyradifurone, no matter whether they cause detrimental metabolic disorders in the bumblebee. These data would shed light on the delicate issue of how these insecticides affect the complex network of gut microbiota–host interactions existing in nature.

## 5. Conclusions

Syrup consumption reduction, egg-laying period delay, and microcolony growth effect data indicated that imidacloprid exposure significantly affected bumblebees. However, flupyradifurone exposure was considered safe. The safety of flupyradifurone was questioned based on the structural and functional profiles of gut microbiota, with obvious upregulation of the lipid and xenobiotic metabolism, and significant suppression of carbohydrate metabolism and membrane transport. We, thus, argue that, even in the case of "safe" appearance, flupyradifurone, similar to imidacloprid, may confront the nAChR-mediated effects through gut microbiota–host interactions. These results, thus, refer to the need to further measure functional metabolic changes within the gut, in response to flupyradifurone exposure, and the potential ramifications of this change to host health, which needs further investigation to clarify its safety.

**Author Contributions:** Conceptualization, Q.Z. and X.W.; methodology, Q.Z. and Q.W.; software, Q.Z.; validation, X.W. and H.Z.; formal analysis, Q.Z.; investigation, Y.Z.; resources, Y.Z and H.Z.; data curation, Q.Z., X.W., Q.W., Y.Z. and H.Z; writing—original draft preparation, Q.Z. and X.W.; writing—review and editing, X.W. and H.Z.; visualization, Y.Z and H.Z.; supervision, Y.Z and H.Z; project administration, H.Z.; funding acquisition, Y.Z. and H.Z. All authors have read and agreed to the published version of the manuscript.

**Funding:** This research was funded by Shandong Provincial Natural Science Foundation (ZR2021YQ21, ZR2021QC218); Shandong Provincial Agriculture Research System (SDAIT-24-01); Shandong Provincial Key R&D Program(2019GHZ028).

**Institutional Review Board Statement:** All animal procedures were conducted according to the ethical principles of the most recent version of the Declaration of Helsinki, and approved by the Ethics Committee of Shandong Academy of Agricultural Sciences (SAAS-2022LL- 01).

**Informed Consent Statement:** Not applicable.

**Data Availability Statement:** Data available in a publicly accessible repository.

**Acknowledgments:** We thank Shandong Institute of Plant Protection for assisting bumblebee in vivo experiments.

**Conflicts of Interest:** The authors declare no conflict of interest.

**Abbreviations**

| | |
|---|---|
| nAchR | nicotinic acetylcholine receptor |
| CTAB | cetyltrimethylammonium bromide |
| PCR | polymerase chain reaction |
| QIIME | quantitative Insights into Microbial Ecology |
| PCOA | principal coordinates analysis |
| PICRUSt | phylogenetic investigation of communities by reconstruction of unobserved states |
| OTU | operational taxonomic units |
| IM-0.1 | 0.1 μg/L imidacloprid |
| IM-1 | 1 μg/L imidacloprid |
| IM-10 | 10 μg/L imidacloprid |
| F-0.1 | 0.1 μg/L flupyradifurone |
| F-1 | 1 μg/L flupyradifurone |
| F-10 | 10 μg/L flupyradifurone |
| CK | control |
| ATP | adenosine triphosphate |
| KEGG | Kyoto Encyclopedia of Genes and Genomes |
| KO | Kyoto Encyclopedia of Genes and Genomes Orthology |

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
