# Peer review of "Impacts of Imidacloprid and Flupyradifurone Insecticides on the Gut Microbiota of Bombus terrestris"

_agriculture, doi:10.3390/agriculture12030389_

Round 1

Reviewer 1 Report

The manuscript focuses on studying the impacts of imidacloprid and flupyradifurone insecticides on the developmental biology and gut microbiota of Bombus terrestris. They found that the highest dose tested, 10μg/L imidacloprid significantly impeded syrup consumption, and postponed the egg-laying period, larvae, and pupae development as well as decreased the gut symbionts whereas 10μg/L flupyradifurone didn’t show any adverse effects on developmental biology and gut microbiota. However, PICRUST analysis of both insecticides showed altered microbial metabolism based on the predicted functional categories. Overall, the study is very interesting and intriguing. The manuscript has been written very well. However, I have some minor concerns:

  1. Figure 2 does not show the data for all the three concentrations of each insecticide tested. It is important to add all the data so it does match with the written results. Full forms of abbreviations used are missing such as F-10, IM-1 etc.
  2. The authors tried to correlate the phenotypic effects with PICRUST analysis. Since both of the insecticides showed similar pattern, irrespective of the phenotypic effects. May be there is no cause-effect relationship between developmental effects and gut microbiota metabolism predicted. These predictions are just based on bioinformatics analysis package, not on the basis of robust data analysis such as transcriptomics or metabolomics. It is important to mention the shortcomings of the study in the discussion section.
  3. The authors observed a significant increase in abundance for Bifidobacterium in bumblebees treated with 10μg/L flupyradifurone and speculated its potential to counteract the chronic effects of the insecticide. Therefore, it is worth testing the effect of this bacteria on Imidacloprid treated bees whether it improves the performance of the bees. This will improve the impact of the study.

Reviewer 2 Report

Impacts of imidacloprid and flupyradifurone insecticides on  the gut microbiota of Bombus terrestris

Summary

In the herein presented manuscript “Impacts of imidacloprid and flupyradifurone insecticides on the gut microbiota of Bombus terrestris” Zhang and colleagues compare the impact of two insecticides targeting the nicotinic acetylcholine receptor on the food consumption, development and gut microbiota of the bumblebee Bombus terrestris.  Previous studies showed that one of the insecticides, imidalcloprid has negative effects on honeybees, whereas the other, flupyradifurone seems to be harmless for honeybees. To investigate the potential different outcomes in bumblebees, despite the apparent same mode of action of the two insecticides, Zhang and colleagues compared the effects on bumblebees when fed with 10 µg/L of imidalcloprid versus those fed with 10 µg/L of flupyradifurone. The authors report differences in food consumption behavior, egg-laying period, larval and pupal development and relative abundance of two bumblebee specific symbionts, Apibacter and Lactobacillis Firm-5.  The authors implemented bioinformatic analysis using the software package PICRUSt to estimate a functional-gene profile for their samples, noting that flupyradifurone, similarly to imidalcloprid, seems to suppress pathways involved in carbohydrate and nucleotide metabolism, translation and membrane transport. Nevertheless, the flupyradifurone seems not to negatively affect colonization of any core member of the bumblebee microbiota. The authors conclude that chronic flupyradifurone exposure may considered safe for bumblebees, noting that the functional profiles deserves further investigation.

General comments

The scientific questions are sound, the experiments were conducted in a proper manner, the manuscript is well written, and the results displayed in the figures are in a clear, concise manner. One thing that should be improved is the size of the sub figures in Fig. 2, they are in my opinion too small, especially the axis descriptions. Since the authors showed with functional profiling later in the manuscript possible negative effects of flupyradifurone, I would suggest toning down the subtitle in 3.2. Chronic exposure to flupyradifurone was considered safe for bumblebees”, into “Chronic exposure to flupyradifurone show no detrimental effects in food uptake and development”. Overall, the topic of the study fits well into the scope of MDPI Agriculture journal, since it addresses the increasingly important question, whether novel insecticides with slight modifications in their target binding sites are less harmless to pollinators such as honeybees and bumblebees and therefore serves as an alternative to the conventional chemical insecticides.

Specific comments

L154 B. terrestris in italics

L327 Remove full stop between Drosophila. melanogaster

Reviewer 3 Report

The topic of this work is very interesting. Research often focuses on the honey bee, but other pollinator bees are also essential for agriculture and therefore important to investigate. The experimental design is well conceived and although there are many suggestions for improvement.

Minor revisions

-Pages 1 and 2 lines 27-28, 28- 30, 65-66, 80-82 bibliographical references are missing from the text. please add them.

-Page 2 line 70 “Bifidobacterium” should be written in italics.

-Page 2 lines 86-88 this part would seem to belong to the 'discussion' section.

-Page 3 line 114 sugar is usually indicated as weight/weight.  Was it only sucrose? In many works you find lower percentages of sugar in the syrup, please quote the work referred to for the bumblebee diet.

-Page 4 line 154 “B.terrestris” should be written in italics.

-Figure 2 the graphs in figure 2 are too small and are unreadable. Please modify them. Please add the meaning of the abbreviations used in the caption. Please add that different letters indicate statistical difference between histogram bars.

-Page 5 lines 175-179 this part would seem to belong to the 'materials and methods' section.

-Page 5 lines 177-178, 178-179 bibliographical references are missing from the text. please add them.

-Page 5 lines 181-183 Please add p-value.

-Page 5 lines 184-186 this part would seem to belong to the 'discussion' section.

-Page 5 lines 191 what does the abbreviation CK mean? please add in the materials and methods the abbreviations used for the different experimental groups.

-Page 6 lines 208-211 this part would seem to belong to the 'materials and methods' section.

-Page 6 lines 213 what does the abbreviation IM-10 mean? please add in the materials and methods the abbreviations used for the different experimental groups.

-Page 6 line 214 how does it differ?

-Page 6 lines 220-222 decreases significantly? Please add p-value.

-Page 6 lines 222-225 this part would seem to belong to the 'discussion' section.

-Figures 3, 5 and 6 Please add the meaning of the abbreviations used in the caption.

-Figures 3 and 4 add theses on the x-axis of the graphs

-Page 8 lines 247-248 this part would seem to belong to the 'discussion' section.

-Page 9-10 lines 266-267, 285-289 these parts would seem to belong to the 'discussion' section.

-Page 11 lines 349-350 bibliographical references are missing from the text. please add them.
